# Compactness of quantics tensor train representations of local imaginary-time propagators

**Haruto Takahashi, Rihito Sakurai and Hiroshi Shinaoka⋆**

Department of Physics, Saitama University, Saitama 338-8570, Japan

⋆ h.shinaoka@gmail.com

## Abstract

Space-time dependence of imaginary-time propagators, vital for *ab initio* and many-body calculations based on quantum field theories, has been revealed to be compressible using Quantum Tensor Trains (QTTs) [Phys. Rev. X 13, 021015 (2023)]. However, the impact of system parameters, like temperature, on data size remains underexplored. This paper provides a comprehensive numerical analysis of the compactness of local imaginary-time propagators in QTT for one-time/-frequency objects and two-time/-frequency objects, considering truncation in terms of the Frobenius and maximum norms. To study worst-case scenarios, we employ random pole models, where the number of poles grows logarithmically with the inverse temperature and coefficients are random. The Green's functions generated by these models are expected to be more difficult to compress than those from physical systems. The numerical analysis reveals that these propagators are highly compressible in QTT, outperforming the state-of-the-art approaches such as intermediate representation and discrete Lehmann representation. For one-time/-frequency objects and two-time/-frequency objects, the bond dimensions saturate at low temperatures, especially for truncation in terms of the Frobenius norm. We provide counting-number arguments for the saturation of bond dimensions for the one-time/-frequency objects, while the origin of this saturation for two-time/-frequency objects remains to be clarified. This paper's findings highlight the critical need for further research on the selection of truncation methods, tolerance levels, and the choice between imaginary-time and imaginary-frequency representations in practical applications.



# 1 Introduction

In modern computational quantum field theory, imaginary-time propagators are pivotal components [1]. Typically, these propagators are multivariate functions, depending on variables such as imaginary-time/-frequency, momentum, and spin-orbital. Notably, the volume of these numerical data escalates, particularly at low temperatures, where a multitude of intriguing phenomena occur. This expansion in data size, coupled with a corresponding increase in computational time, constrains the practical application of advanced quantum field theory calculations.

Recent developments in quantum field theory, particularly at the two-particle level [2–6], have garnered significant attention for their capability to characterize dynamical responses to external fields and to address non-local strong correlation effects. These theories are based on multi-particle propagators, for instance, two-particle Green's functions, which are dependent on several frequencies/times. Consequently, the development of a succinct representation for these multi-particle propagators has emerged as a critical challenge in the field of condensed matter physics. This has led to considerable efforts in formulating compact representations for both one-particle [7–11] and two-particle propagators [12–15].

A novel proposition in this realm is the application of the so-called quantics tensor train (QTT) [16–18] for the compression of space-time dependence in propagators [19]. QTTs have been widely attractive for their ability to compress high-dimensional data in various fields of natural science by exploiting the separation of length scales such as turbulence [20,21], plasma physics [22], and quantum chemistry [23]. This approach compresses general space-time dependencies, encompassing imaginary-time/-frequency, real-time/-frequency, and momentum, through the use of tensor train (TT). The core principle of this compression technique lies in separation in length scale, a phenomenon anticipated to be widespread in various contexts. The QTT method is particularly noteworthy for its generality and applicability beyond just imaginary-time propagators. It also facilitates a range of efficient operations, such as convolution and (quantum) Fourier transform, among others. Previous studies have provided numerical evidence supporting the compressibility of data derived from diverse quantum field calculations. However, a comprehensive understanding of the fundamental attributes of the QTT representation, such as the scaling of data size with temperature, remains largely unexplored.

In this paper, we study the data size dependence of QTT representations of one-time/-frequency objects and two-time/-frequency objects. We use *maximally* random pole models to examine the compactness of QTTs in worst-case scenarios. We explore two distinct truncation approaches based on the Frobenius and maximum norms.

The paper is organized as follows. Section II is devoted to introducing QTT and manipulations of TTs. We show results for one-time/-frequency objects in Sec. III, and two-time/-frequency objects in Sec. IV. The summary and conclusion are given in Sec. V.

## 2 Quantics representation

We introduce QTT and operations with TTs.

### 2.1 Quantics tensor train (QTT)

We briefly introduce the efficient discretization and data compression of a scalar-valued function using QTT [16–18]. First, we consider a function $f(x)$ defined on $x \in [0,1)$ and discretize it by setting up a grid of $2^{\mathcal{R}}$ equally spaced points on the $x$ axis. The resolution can be exponentially improved with respect to the value of $\mathcal{R}$, but the amount of data also increases exponentially. We represent the variable $x$ in binary form $x = (0.\sigma_1\sigma_2\cdots\sigma_{\mathcal{R}})_2$ and separates it into $\mathcal{R}$ exponentially different length scales. Then, the discretized $f(x)$ is reshaped into an $\mathcal{R}$-way tensor:

$$f(x) = f(\sigma_1, \sigma_2, \cdots, \sigma_{\mathcal{R}}) = F_{\sigma_1\sigma_2\ldots\sigma_{\mathcal{R}}}. \tag{1}$$

Furthermore, this $\mathcal{R}$-way tensor $F_{\sigma_1\sigma_2\ldots\sigma_{\mathcal{R}}}$ is approximated by a TT using decomposition methods such as singular value decomposition (SVD) and tensor cross interpolation (TCI) [24–27]:

$$F_{\sigma_1\sigma_2\ldots\sigma_{\mathcal{R}}} \approx \sum_{\alpha_1=1}^{\chi_1} \cdots \sum_{\alpha_{\mathcal{R}-1}=1}^{\chi_{\mathcal{R}-1}} F^{(1)}_{\sigma_1,1\alpha_1} \cdots F^{(l)}_{\sigma_l,\alpha_{l-1}\alpha_l} \cdots F^{(\mathcal{R})}_{\sigma_{\mathcal{R}},\alpha_{\mathcal{R}-1}1}$$
$$\equiv F^{(1)}_{\sigma_1} \cdot (\cdots) \cdot F^{(l)}_{\sigma_l} \cdot (\cdots) \cdot F^{(\mathcal{R})}_{\sigma_{\mathcal{R}}}. \tag{2}$$

Here, $F^{(l)}_{\sigma_l}$ is a three-way tensor of dimensions $2 \times \chi_{l-1} \times \chi_l$, $\alpha_l$ represents the $l$-th virtual bond, and $\chi_l$ represents the bond dimension. The original data volume of $2^{\mathcal{R}}$ can be compressed to $O(\chi^2\mathcal{R})$, provided $\chi \ll 2^{\mathcal{R}/2}$ ($\chi$ is the maximum value of $\chi_l$).

There are several analytic functions whose explicit low-rank QTT representations are known. A simple example is the exponential function $g(x) = e^{ax}$ ($a$ is a constant). In the binary form, $g(x)$ can be represented as

$$g(x) = e^{ax} = e^{a\sigma_1/2} \cdot e^{a\sigma_2/2^2} \cdots e^{a\sigma_{\mathcal{R}}/2^{\mathcal{R}}}. \tag{3}$$

This indicates that the exact QTT representation of $g(x)$ has a bond dimension of 1. As will be explained later, the imaginary-time Green's function $G(\tau)$ can be expressed as the sum of exponential functions.

## 2.2 Scaling a TT by a scalar

The product of a given TT, $F_{\text{TT}}$, and a constant $c$ can be computed by multiplying one core tensor of $F_{\text{TT}}$ by $c$ as

$$F'_{\sigma_1\sigma_2\ldots\sigma_{\mathcal{R}}} = \sum_{\alpha_1=1}^{\chi_1} \cdots \sum_{\alpha_{\mathcal{R}-1}=1}^{\chi_{\mathcal{R}-1}} (cF^{(1)}_{\sigma_1,1\alpha_1})F^{(2)}_{\sigma_2,\alpha_1\alpha_2} \cdots F^{(\mathcal{R})}_{\sigma_{\mathcal{R}},\alpha_{\mathcal{R}-1}1}, \tag{4}$$

which does not change the bond dimension.

## 2.3 Adding TTs

We explain how to add multiple TTs without a truncation. First, we define two TTs $A_{\text{TT}}$ and $B_{\text{TT}}$:

$$A_{\text{TT}} = \sum_{\alpha_1,\cdots,\alpha_{\mathcal{R}-1}=1}^{\chi_A} A^{(1)}_{\sigma_1,1\alpha_1} A^{(2)}_{\sigma_2,\alpha_1\alpha_2} A^{(3)}_{\sigma_3,\alpha_2\alpha_3} \cdots A^{(\mathcal{R})}_{\sigma_{\mathcal{R}},\alpha_{\mathcal{R}-1}1}$$
$$\equiv A^{(1)}_{\sigma_1} \cdot A^{(2)}_{\sigma_2} \cdot A^{(3)}_{\sigma_3} \cdot (\cdots) \cdot A^{(\mathcal{R})}_{\sigma_{\mathcal{R}}}, \tag{5}$$

$$B_{\text{TT}} = \sum_{\beta_1,\cdots,\beta_{\mathcal{R}-1}=1}^{\chi_B} B^{(1)}_{\sigma_1,1\beta_1} B^{(2)}_{\sigma_2,\beta_1\beta_2} B^{(3)}_{\sigma_3,\beta_2\beta_3} \cdots B^{(\mathcal{R})}_{\sigma_{\mathcal{R}},\beta_{\mathcal{R}-1}1}$$
$$\equiv B^{(1)}_{\sigma_1} \cdot B^{(2)}_{\sigma_2} \cdot B^{(3)}_{\sigma_3} \cdot (\cdots) \cdot B^{(\mathcal{R})}_{\sigma_{\mathcal{R}}}. \tag{6}$$

For simplicity, we assumed that the sizes of the virtual bonds of $A_{\text{TT}}$ and $B_{\text{TT}}$ are $\chi_A$ and $\chi_B$, respectively. The TT representing their sum, $C_{\text{TT}}(= A_{\text{TT}} + B_{\text{TT}})$ can be constructed explicitly as

$$C_{\text{TT}} = A^{(1)}_{\sigma_1} \cdot A^{(2)}_{\sigma_2} \cdot A^{(3)}_{\sigma_3} \cdot (\cdots) \cdot A^{(\mathcal{R})}_{\sigma_{\mathcal{R}}} + B^{(1)}_{\sigma_1} \cdot B^{(2)}_{\sigma_2} \cdot B^{(3)}_{\sigma_3} \cdot (\cdots) \cdot B^{(\mathcal{R})}_{\sigma_{\mathcal{R}}}$$
$$= \begin{pmatrix} A^{(1)}_{\sigma_1} & B^{(1)}_{\sigma_1} \end{pmatrix} \begin{pmatrix} A^{(2)}_{\sigma_2} & 0 \\ 0 & B^{(2)}_{\sigma_2} \end{pmatrix} \begin{pmatrix} A^{(3)}_{\sigma_3} & 0 \\ 0 & B^{(3)}_{\sigma_3} \end{pmatrix} \cdots \begin{pmatrix} A^{(\mathcal{R})}_{\sigma_{\mathcal{R}}} \\ B^{(\mathcal{R})}_{\sigma_{\mathcal{R}}} \end{pmatrix}$$
$$\equiv C^{(1)}_{\sigma_1} \cdot C^{(2)}_{\sigma_2} \cdot C^{(3)}_{\sigma_3} \cdot (\cdots) \cdot C^{(\mathcal{R})}_{\sigma_{\mathcal{R}}}. \tag{7}$$

From Eq. (7), one can see that the bond dimension of $C_{\text{TT}}$ is $\chi_C = \chi_A + \chi_B$, indicating that the bond dimensions are additive in the case of exact TT summation. However, the resultant TT may contain redundancies. This can be understood by considering $A + A = 2A$, where the bond dimension should not change. Such redundancies can be removed by truncating the TT in terms of some norm as we will see below.

## 2.4 Truncation schemes

Redundancies in a TT can be eliminated by truncating bond dimensions either in terms of the Frobenius norm or the maximum norm. Below, we will explain these two truncation schemes.

### 2.4.1 Truncation in terms of the Frobenius norm

By using SVD, the original TT, $F_{\mathrm{TT}}$, can be approximated by a TT with smaller (optimized) bond dimensions $\tilde{F}_{\mathrm{TT}}$ so that

$$\frac{\|F_{\mathrm{TT}} - \tilde{F}_{\mathrm{TT}}\|_{\mathrm{F}}^2}{\|F_{\mathrm{TT}}\|_{\mathrm{F}}^2} < \epsilon_{\mathrm{F}}, \tag{8}$$

where $\|\cdots\|_{\mathrm{F}}$ denotes the Frobenius norm. Note that following common convention, the approximation error is expressed as a squared deviation. For a given tolerance $\epsilon_{\mathrm{F}}$, the SVD-based truncation yields a low-rank TT approximation that is optimal with the smallest rank. We refer the reader to Ref. [28] for more technical details.

### 2.4.2 Truncation in terms of the maximum norm

The bond dimensions of a TT can be truncated in terms of the maximum norm by TCI. TCI is a heuristic method to construct a low-rank TT representation of a given tensor, $F_{\sigma_1, \cdots, \sigma_{\mathcal{L}}}$ [24–27]. TCI does not require knowing all the elements in the target tensor $F_{\sigma_1, \cdots, \sigma_{\mathcal{L}}}$ but only requires the ability to evaluate the tensor at any given index $(\sigma_1, \cdots, \sigma_{\mathcal{L}})$. Thus, the tensor can be implemented as a function that computes its value on the fly at a given index. TCI constructs a TT using the values of the function evaluated on adaptively chosen interpolation points (*pivots*), minimizing the error in the maximum norm

$$\epsilon_{\max} = \|F_{\mathrm{TT}} - \tilde{F}_{\mathrm{TT}}\|_{\max}, \tag{9}$$

where $\tilde{F}_{\mathrm{TT}}$ is a low-rank approximation. The number of function evaluations required to construct $\tilde{F}_{\mathrm{TT}}$ of bond dimension $\chi$ is roughly proportional to the number of parameters in $\tilde{F}_{\mathrm{TT}}$ and scales as $O(\chi^2 \mathcal{L})$. TCI is particularly useful when the target tensor is too large to be stored in memory. We refer the reader to Refs. [27, 29] for more technical details on TCI and its combination with QTT.

## 3 Results: One-time/-frequency objects

### 3.1 Maximally random pole model

We consider the fermionic random pole model defined by

$$G^{\mathrm{F}}(\tau) = \sum_{p=1}^{L} c_p \frac{e^{-\tau w_p}}{1 + e^{-\beta w_p}}, \tag{10}$$

for the interval $0 \leq \tau < \beta$. The pole set $W = \{w_1, \cdots, w_L\}$ is ascertained through the Discrete Lehmann Representation (DLR) [10]. In practical implementations, the intermediate representation (IR) [8] grid is employed [11]. The DLR's size, represented by $L$, increases logarithmically as $O(\log(\beta \omega_{\max}) \log(1/\epsilon))$ [10], where $\omega_{\max}$ is an ultraviolet cutoff, $\beta$ represents the inverse temperature, and $\epsilon$ is a cutoff parameter that dictates the precision of the representation. The distribution of $w_p$ is roughly logarithmic, characterized by a higher density near $\omega = 0$. These positions are spread within the interval $[-\omega_{\max}, \omega_{\max}]$ non-uniformly. The logarithmic distribution of $w_p$ enables a discrete spectrum model to encompass the physical Green's function space, which is characterized by a continuous spectral function defined by the parameters $\beta$ and $\omega_{\max}$. Throughout this study, we take $\omega_{\max} = 1$ and $\epsilon = 10^{-15}$, and $L$ ranges from 19 ($\beta = 10$) to 202 ($\beta = 10^7$). We confirmed that increasing the number of poles in the model does not qualitatively change the results presented below. Since the pole

(a) 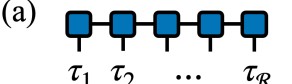

(b) 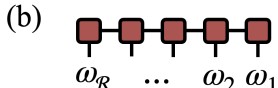

Figure 1: QTT representations for $G^{\mathrm{F}}(\tau)$ [(a)] and $G^{\mathrm{F}}(\mathrm{i}\omega)$ [(b)]. The $\tau_1$ and $\omega_{\mathcal{R}}$ correspond to the largest time and smallest frequency scales, respectively.

distribution is dense enough, we do not randomize the pole positions. The coefficients $c_p$ are independently sampled from a uniform distribution on $[0, 1]$, and are then normalized such that $\sum_p c_p = 1$ ($c_p$ are non-negative). As shown in Appendix A, using a uniform distribution on $[-1/2, 1/2]$, where $c_p$ can be positive or negative, does not change our results qualitatively.

Defining $x \equiv \tau/\beta$, the contribution of each pole $e^{-\tau w_p}$ to $G^{\mathrm{F}}(\tau)$ is represented by a TT of bond dimension 1. As a result, the exact QTT representation of Eq. (10) has a bond dimension of $L \propto \log \beta$. Nevertheless, the exact QTT is redundant and can be compressed significantly numerically, as we will see later.

Additionally, the model's corresponding imaginary-frequency representation is formulated as:

$$G^{\mathrm{F}}(\mathrm{i}\omega_n) = \sum_{p=1}^{L} \frac{c_p}{\mathrm{i}\omega_n - w_p}\,, \tag{11}$$

where $\omega_n = (2n+1)\pi/\beta$ denotes a fermionic imaginary frequency.

In this section, we restrict our analyses to the fermionic cases because the fermionic Green's function can contain more information than the bosonic one [30]. We show results for a bosonic model in Appendix B. We obtained qualitatively the same results as the fermionic cases.

## 3.2 QTT representations

Figure 1 illustrates the QTT representations for the imaginary-time/-frequency objects. The bit representation for $\tau$ is defined as $\tau/\beta = (0.\tau_1 \cdots \tau_{\mathcal{R}})_2$. The bit representation for the imaginary-frequency domain is defined as $n + 2^{\mathcal{R}-1} = (\omega_1 \cdots \omega_{\mathcal{R}})_2$, where $\omega_n = (2n+1)\pi/\beta$ for fermions and $\omega_n = 2n\pi/\beta$ for bosons. The values of $n + 2^{\mathcal{R}-1}$ run from 0 to $2^{\mathcal{R}} - 1$. Notably, the most significant (most left) bit $\tau_1$ corresponds to the most extended length scale in time, whereas $\omega_{\mathcal{R}}$ signifies the smallest length scale in frequency. These two bits are linked via the Fourier transform.

## 3.3 Results for $G^{\mathrm{F}}(\tau)$

Let us start our analyses with $G^{\mathrm{F}}(\tau)$. We construct QTTs truncated in terms of the Frobenius norm as follows. For each random sample, we add the QTTs of the $L$ exponential functions with the coefficients $c_p$, and truncate the result with a desired $\epsilon_{\mathrm{F}}$ by SVD. On the other hand, we construct a QTT truncated in terms of the maximum norm by TCI, which requires only evaluating $G^{\mathrm{F}}(\tau)$ on adaptively chosen interpolation points.

Figures 2 (a) and 2(b) summarize results obtained with the two truncation schemes, respectively. We first discuss the results for the truncation in terms of the Frobenius norm shown in Fig. 2 (a). The left panel of Fig. 2(a) compares the original data and the reconstructed one from the TT truncated with $\epsilon_{\mathrm{F}} = 10^{-10}$. The absolute error is as small as $10^{-6}$ around $\tau = \beta/2$ and shows some increase around $\tau = 0$ and $\beta$, which is ascribed to the large values of the original data around these points.

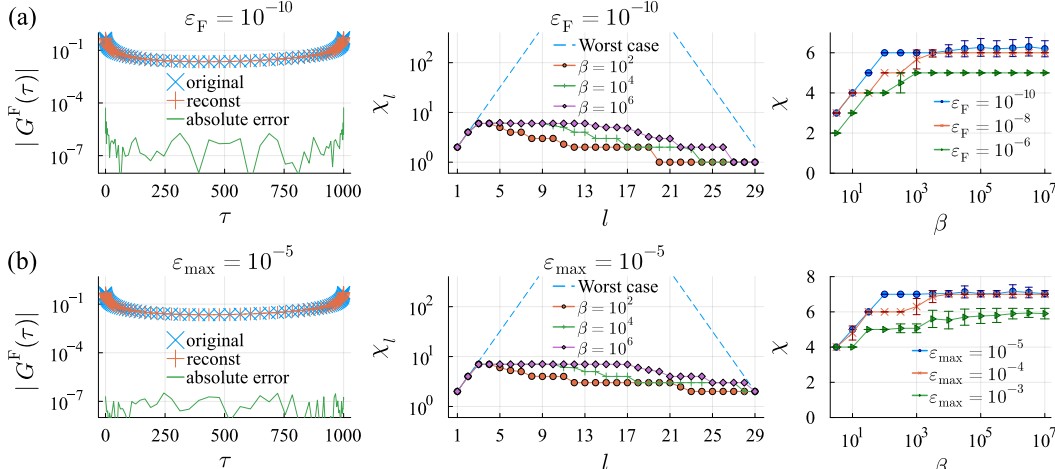

Figure 2: Results for $G^{\mathrm{F}}(\tau)$: (a) with $\epsilon_{\mathrm{F}}$, (b) with $\epsilon_{\max}$. The left panels compare the original (reference) and reconstructed data of a typical sample for $\beta = 10^3$ and $\mathcal{R} = 30$ [the same sample is shown in (a) and (b)]. The middle and right panels show sample-averaged bond dimensions with 30 samples. The broken lines in the middle panels denote the worst exponential growth of $\chi_l$ for incompressible data.

The middle panel of Fig. 2(a) shows the bond dependence of $\chi_l$ for various values of $\beta$. The $\chi_l$ shows an exponential growth at the first few bonds, followed by a slow decay (*tail*) to 1. The length of the tail is determined by the minimum number of bits required to resolve the smallest features in $G^{\mathrm{F}}(\tau)$ ($\equiv \mathcal{R}_{\min}$), which scales as $\mathcal{R}_{\min} \propto \log(\beta \omega_{\max})$.

As shown in the right panel of Fig. 2(a), the maximum value of $\chi_l$, i.e., the bond dimension of the TTs, $\chi$, shows a logarithmic growth at small $\beta$ and becomes saturated at large $\beta$ irrespective of $\epsilon_{\mathrm{F}}$. The weak dependence of $\chi$ at small $\beta$ is consistent with the observation in the previous study [19]. The saturation of $\chi$ is a new finding revealed by our systematic analyses at low temperatures. The saturated value of $\chi$ is substantially smaller than $L$ (the number of poles, $L = 202$ at $\beta = 10^7$), indicating the exact QTT of $G^{\mathrm{F}}(\tau)$ is highly compressible. With increasing $\epsilon_{\mathrm{F}}$, the onset of the saturation shifts to larger $\beta$, and the saturated value of $\chi$ becomes smaller. As we will see later, this logarithmic growth of $\chi$ at small $\beta$ is characteristic to $G^{\mathrm{F}}(\tau)$ and does not exist for $G^{\mathrm{F}}(i\omega)$.

The saturation may be explained by counting-number arguments. The number of elements in a QTT representation is proportional to $\chi^2 \mathcal{R}_{\min}$. On the other side, the number of imaginary-time/-frequency points required to sample the propagator is proportional to $\log(\beta \omega_{\max})$ in the intermediate representation (IR) [8, 11] and DLR [10] grids. By comparing the two, we can conclude that the bond dimension $\chi$ is constant at large $\beta \omega_{\max}$.

Another possible explanation for the saturation of $\chi$ may be provided by a multiscale interpolative construction of QTTs [31]. This mathematical framework allows for the construction of a QTT for a univariate function with a fixed bond dimension, if the function has a multiscale structure; i.e., at each scale, the function can be approximated by Chebyshev interpolation with a fixed number of nodes except a few "dangerous" subintervals. For the Green's function, this condition may be satisfied because the function is difficult to approximate only around $\tau = 0$ and $\beta$.

Figure 2(b) shows results obtained with the truncation using $\epsilon_{\max}$. The results are qualitatively similar to those obtained with the truncation in the Frobenius norm. This indicates the robustness of the aforementioned finding (e.g., the saturation of $\chi$). One minor difference from the truncation in the Frobenius norm is that the absolute error depends on $\tau$ more weakly when truncated in the maximum norm [see the left panel of Fig. 2(b)].

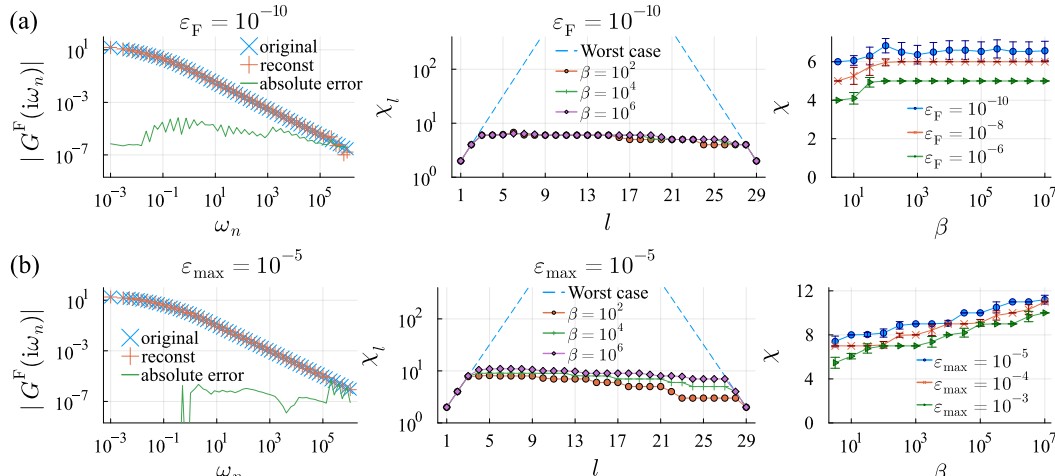

Figure 3: Results for $G^{\text{F}}(i\omega)$: (a) with $\epsilon_{\text{F}}$, (b) with $\epsilon_{\text{max}}$. The left panels compare the original (reference) and reconstructed data of a typical sample for $\beta = 10^3$ and $\mathcal{R} = 30$ [the same sample is shown in (a) and (b)]. The middle and right panels show sample-averaged bond dimensions with 30 samples. The broken lines in the middle panels denote the worst exponential growth of $\chi_l$ for incompressible data.

## 3.4 Results for $G^{\text{F}}(i\omega)$

We now move on to the analyses for $G^{\text{F}}(i\omega)$. Figures 3 (a) and 3(b) show results obtained by the truncation with $\epsilon_{\text{F}}$ and $\epsilon_{\text{max}}$, respectively. For the truncation in terms of the Frobenius norm, we first construct a TT for $G^{\text{F}}(\tau)$ and transform it to the imaginary-frequency domain by the Fourier transform within the QTT representation [19, 32, 33], and truncate the result with a desired $\epsilon_{\text{F}}$ by SVD. On the other hand, for the truncation in the maximum norm, we construct a TT for $G(i\omega)$ directly with TCI. The left panels of Figs. 3(a) and 3(b) compare the original $G^{\text{F}}(i\omega)$ and a reconstructed one for a typical sample, showing a good agreement.

The middle panels of Fig. 3 display the distribution of the bond dimension $\chi_l$. As illustrated in Fig. 1, the leftmost TT index corresponds to the smallest frequency scale, which is linked to the largest time scale. The exponential increase at the leftmost indices is thus consistent with our observation for $G^{\text{F}}(\tau)$. However, $\chi_l$ exhibits a slower decay at large $l$ than $G^{\text{F}}(\tau)$. This may be because there is no trivial QTT representation of bond dimension 1 for $1/i\omega$.

As seen in the right panels, for both truncation schemes, $\chi$ weakly depends on $\beta$ and does not show a logarithmic growth at small $\beta$ in contrast to $G^{\text{F}}(\tau)$. Note that $\chi$ mildly grows with $\beta$ for the truncation in terms of the maximum norm [Fig. 3(b)]. This may be because the absolute maximum value of $G^{\text{F}}(i\omega)$ increases linearly with $\beta$ while the cutoff $\epsilon_{\text{max}}$ is fixed.

The Fronbenius norm is conserved in the Fourier transform between $\tau$ and $i\omega$, which allows us a direct comparison of $\chi$'s for $G^{\text{F}}(\tau)$ and $G^{\text{F}}(i\omega)$ obtained with the same $\epsilon_{\text{F}}$. Comparing the right panels of Fig. 2(a) and Fig. 3(a), one can conclude that the saturated value of $\chi$ at large $\beta$ is almost the same for both representations. This indicates that the imaginary-time representation is more favorable regarding data size at small $\beta$ and large $\epsilon_{\text{F}}$.

## 3.5 Computational complexity

In concluding this section, we discuss the computational complexity inherent in the principal operations of the Green's function within the QTT. The empirical evidence presented herein strongly suggests the saturation of $\chi$ at large $\beta$ when the cutoff remains fixed. Conversely, the bond dimension of the Matrix Product Operator (MPO) representing the Fourier trans-

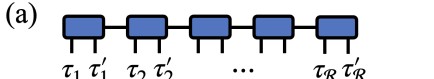
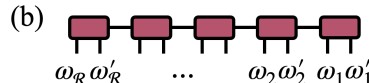

Figure 4: QTT representations for $G(\tau, \tau')$ [(a)] and $G(i\omega_n, i\omega_{n'})$ [(b)].

form, which is as $\chi_{\mathrm{QFT}}$, is independent of the number of bits $\mathcal{R}$ for a fixed $\epsilon_{\mathrm{F}}$ [19, 32, 33] or $\epsilon_{\max}$ [27]. These indicate that the computational complexity associated with the Fourier transform of the Green's function scales as $O(\chi^3 \chi_{\mathrm{QFT}}^3 \mathcal{R}) = O(\log(\beta \omega_{\max}))$ at low temperatures. This computational efficiency is anticipated to asymptotically surpass traditional methodologies based on IR [34] or DLR [10]. In these conventional frameworks, the Fourier transform necessitates a matrix-vector multiplication of size $L$, incurring a computational expense of $O(L^2)$, where $L \propto \log(\beta \omega_{\max})$. Nevertheless, given the substantial overhead associated with MPO-MPO contractions, the Fourier transform within the QTT framework may offer advantages only when supplementary degrees of freedom, for instance, momentum dependence, are encapsulated within a TT.

# 4 Results: Two-time/-frequency objects

In this section, we investigate QTT representations of two-frequency/-time objects.

## 4.1 Maximally random pole model

We consider two models defined in the fermion-fermion and fermion-boson channels. We analyze these two models separately because their superposition only sums up the bond dimensions of the two models (see Sec. 2.3), not affecting scaling behaviors with $\beta$. We do not consider a constant term in the Matsubara frequency space and anomalous propagator terms at zero bosonic frequency because these contributions can be represented by a single TT of bond dimension 1.

### 4.1.1 Fermion-fermion model

The fermion-fermion model is defined by

$$G^{\mathrm{FF}}(i\omega_n, i\omega_{n'}) = \sum_{p=1}^{L} \sum_{p'=1}^{L} \frac{c_{pp'}}{(i\omega_n - w_p)(i\omega_{n'} - w_{p'})}, \tag{12}$$

where we used $\omega_{\max} = 1$ and $\epsilon = 10^{-10}$ for generating the poles. we use the same pole positions as those used for $G^{\mathrm{F}}(i\omega)$. The Green's function depends on two fermionic frequencies. The coefficients $c_{pp'}$ are independently sampled from a uniform distribution on $[0, 1]$, and are then normalized such that $\sum_{pp'} c_{pp'} = 1$. The model can be transformed to the imaginary-time domain as

$$G^{\mathrm{FF}}(\tau, \tau') = \sum_{p=1}^{L} \sum_{p'=1}^{L} c_{pp'} U^{\mathrm{F}}(\tau; w_p) U^{\mathrm{F}}(\tau'; w_{p'}), \tag{13}$$

where $U^{\mathrm{F}}(\tau; w) = -e^{-\tau w}/(1 + e^{-\beta w})$ $(0 \leq \tau \leq \beta)$ and $U^{\mathrm{F}}(\tau; w) = -U^{\mathrm{F}}(\tau + \beta; w)$.

### 4.1.2 Fermion-boson model

The fermion-boson model is defined by

$$G^{\text{FB}}(\mathrm{i}\omega_n, \mathrm{i}\omega_{n'}) = \sum_{p=1}^{L} \sum_{p'=1, w_{p'} \neq 0}^{L} \frac{w_{p'} c_{pp'}}{(\mathrm{i}\omega_n - w_p)(\mathrm{i}\omega_{n'} - \mathrm{i}\omega_n - w_{p'})}, \tag{14}$$

where we used $\omega_{\text{max}} = 1$ and $\epsilon = 10^{-10}$ for generating the poles and introduced the linear constraint $w_{p'} \neq 0$ to avoid the divergence at $\mathrm{i}\omega_{n'} - \mathrm{i}\omega_n = 0$. The Green's function depends on a fermionic frequency $\mathrm{i}\omega_n$ and a bosonic frequency $\mathrm{i}\omega_{n'} - \mathrm{i}\omega_n$. The Green's function can be transformed to the imaginary-time domain as

$$G^{\text{FB}}(\tau, \tau') = \sum_{p=1}^{L} \sum_{p'=1, w_{p'} \neq 0}^{L} w_{p'} c_{pp'} U^{\text{F}}(\tau; w_p) U^{\text{B}}(\tau' - \tau; w_{p'}), \tag{15}$$

where $U^{\text{B}}(\tau; w) = -e^{-\tau w}/(1 - e^{-\beta w})$ $(0 < \tau < \beta)$ and $U^{\text{B}}(\tau; w) = U^{\text{B}}(\tau + \beta; w)$. Note that $G^{\text{FB}}(\tau, \tau')$ exhibits a line discontinuity at $\tau = \tau'$.

## 4.2 QTT representations

Figure 4 illustrates QTT representations for the two-time/-frequency objects. The two physical (non-virtual) indices of each TT tensor correspond to the same length scale. This is favorable because these degrees of freedom are entangled [19]. The virtual bonds correcting two neighboring tensors carry entanglement between the different scales.

When performing TCI or SVD, we fuse the two indices at each tensor as a single index of dimension 4. We contract a TT truncated in terms of the Frobenius norm as follows. We first construct a TT for $G(\tau, \tau')$ or $G(\mathrm{i}\omega_n, \mathrm{i}\omega_{n'})$ by TCI with a sufficiently small $\epsilon_{\text{max}}$ and then truncate it with a desired $\epsilon_{\text{F}}$ by SVD. On the other hand, we construct a TT truncated in terms of the maximum norm for a desired $\epsilon_{\text{max}}$ directly by TCI.

## 4.3 Results: Imaginary time

### 4.3.1 Fermion-fermion model

Figure 5 summarizes results obtained for $G^{\text{FF}}(\tau, \tau')$ by the truncation using $\epsilon_{\text{F}}$ [(a)] and $\epsilon_{\text{max}}$ [(b)] with $\mathcal{R} = 40$. For both in Figs. 5(a) and 5(b), $\chi_l$ converges to $O(1)$ at large $l$, indicating that $\mathcal{R} = 40$ is large enough. For the two truncation schemes, the bond dimension $\chi$ grows at small $\beta$ and saturates at large $\beta$. The onset of the saturation shifts to larger $\beta$ as $\epsilon_{\text{F}}$ or $\epsilon_{\text{max}}$ is reduced. This saturation is a surprising finding, considering the following naive counting-number arguments. The number of elements in a QTT representation is proportional to $\chi^2 \mathcal{R}_{\text{min}}$, while the number of coefficients in $c_{pp'}$ is proportional to $L^2 \propto (\log(\beta \omega_{\text{max}}))^2$. By comparing the two, we may conclude that the bond dimension $\chi \propto (\log(\beta \omega_{\text{max}}))^{1/2}$ at large $\beta$, which is apparently inconsistent with the observed saturation. The multiscale interpolative construction of QTTs [31] may not provide a direct explanation either, as the function is difficult to approximate not only around isolated points $(\tau, \tau') = (0, 0)$ and $(\beta, \beta)$, but also rectangular areas near $\tau = 0, \beta$ and $\tau' = \beta$. A comprehensive understanding of the saturation of $\chi$ at large $\beta$ remains an open question.

### 4.3.2 Fermion-boson model

Figure 6 shows results obtained for $G^{\text{FB}}(\tau, \tau')$, which exhibits a line discontinuity at $\tau = \tau'$. For the truncation in terms of the Frobenius norm, the error is slightly enhanced near the diagonal

line, which can be attributed to the enhanced values of $|G^{\mathrm{FB}}(\tau, \tau')|$. As seen in Fig. 6(b), this is suppressed by using the truncation in the maximum norm, indicating the robustness of the QTT representation for the line discontinuity.

A remarkable finding is that $\chi$ reaches a maximum and shows even a slow decay at large $\beta$. This clearly indicates that $G^{\mathrm{FB}}(\tau, \tau')$ contains less information than $G^{\mathrm{FF}}(\tau, \tau')$. This can be ascribed to the fact that the imaginary-time dependence of a bosonic propagator includes less information than the fermionic one [30]. This fact can be understood in the following manner. The spectral function of a bosonic one-particle propagator vanishes linearly around the zero frequency, indicating the absence of low-energy excitations. This is transformed to the imaginary-time domain as localized structures near $\tau = 0$ and $\beta$, whose description requires a smaller bond dimension than extended ones. Although the above argument is based on single-frequency objects, it is expected to hold for two-frequency objects as well. To be more specific, two-frequency objects have a similar structure concerning bosonic frequency dependence [see $w_{p'}$ in the numerator of Eq. (14)]. This may explain the smaller bond dimension of $G^{\mathrm{FB}}(\tau, \tau')$ compared to $G^{\mathrm{FF}}(\tau, \tau')$.

## 4.4 Results: Imaginary frequency

### 4.4.1 Fermion-fermion model

Figure 7 summarizes results obtained for $G^{\mathrm{FF}}(i\omega_n, i\omega_{n'})$ with the truncation in terms of $\epsilon_{\mathrm{F}}$ [(a)] and $\epsilon_{\max}$ [(b)] with $\mathcal{R} = 40$. Note that the maximum norm of the Green's function diverges toward zero temperature.

Let us first discuss the results for the truncation with $\epsilon_{\mathrm{F}}$ shown in Fig. 7(a). Comparing Fig. 7(a) with Fig. 5(a), one can see that the bond dimension $\chi$ is smaller for $G^{\mathrm{FF}}(\tau, \tau')$ than $G^{\mathrm{FF}}(i\omega_n, i\omega_{n'})$ at small $\beta$ for the same cutoff $\epsilon_{\mathrm{F}}$. This indicates that the imaginary-time representation is more favorable regarding data size at low temperatures, similar to the case of the one-time/-frequency objects. As seen in Fig. 7(a), $\chi$ seems to saturate at large $\beta$. Similarly to the one-time/-frequency objects, the onset of the saturation shifts to larger $\beta$ as $\epsilon_{\mathrm{F}}$ is reduced. One notable observation is that $\chi$ decreases slightly around $\beta = 10^6$ for the smallest $\epsilon_{\mathrm{F}} = 10^{-8}$. Its origin is unclear and requires further investigation in future studies.

We now move on to the results for the truncation with $\epsilon_{\max}$ shown in Fig. 7(b). For the truncation in terms of the maximum norm, as seen in Fig. 7(b), the bond dimension $\chi$ for $G^{\mathrm{FF}}(i\omega_n, i\omega_{n'})$ diverges at large $\beta$. This may be because the maximum norm of the Green's function diverges and the fixed cutoff $\epsilon_{\max}$ becomes too strict at low temperatures. Increasing $\epsilon_{\max}$ reduces $\chi$ overall, but the $\beta$ dependence of $\chi$ remains unchanged qualitatively.

An appropriate choice of $\epsilon_{\mathrm{F}}$ and $\epsilon_{\max}$ at low temperatures may be crucial for practical calculations. This is expected to depend on the model, the observable, and the computation to be performed. Further investigations are required in future case-by-case studies.

### 4.4.2 Fermion-boson model

Figure 8 shows the results for $G^{\mathrm{FB}}(i\omega_n, i\omega_{n'})$. The $G^{\mathrm{FB}}(i\omega_n, i\omega_{n'})$ exhibits a slowly decaying tail at the diagonal line. The bond dimension $\chi_l$ shows a similar behavior to that of $G^{\mathrm{FF}}(i\omega_n, i\omega_{n'})$ regardless of the truncation schemes. A notable difference from the results for $G^{\mathrm{FF}}(i\omega_n, i\omega_{n'})$ is that $\chi_l$ converges to a higher value at large $\beta$. This is likely due to the diagonal structure. However, this does not matter in practice because the grid size can be exponentially large with $\mathcal{R}$, and the effects of truncation at high frequencies are expected to become exponentially small. Overall, these results establish the robustness of the QTT framework for the two-time/-frequency objects with diagonal structures. Similarly to the fermion-fermion model, $\chi$ is smaller for $G^{\mathrm{BB}}(\tau, \tau')$ than $G^{\mathrm{BB}}(i\omega_n, i\omega_{n'})$ at small $\beta$ for the same cutoff $\epsilon_{\mathrm{F}}$. Furthermore, $\chi$ is smaller for $G^{\mathrm{FB}}(i\omega_n, i\omega_{n'})$ than $G^{\mathrm{FF}}(i\omega_n, i\omega_{n'})$ at large $\beta$ for the same cutoff $\epsilon_{\max}$, being consistent with that $G^{\mathrm{FB}}(\tau, \tau')$ contain less information than $G^{\mathrm{FF}}(\tau, \tau')$.

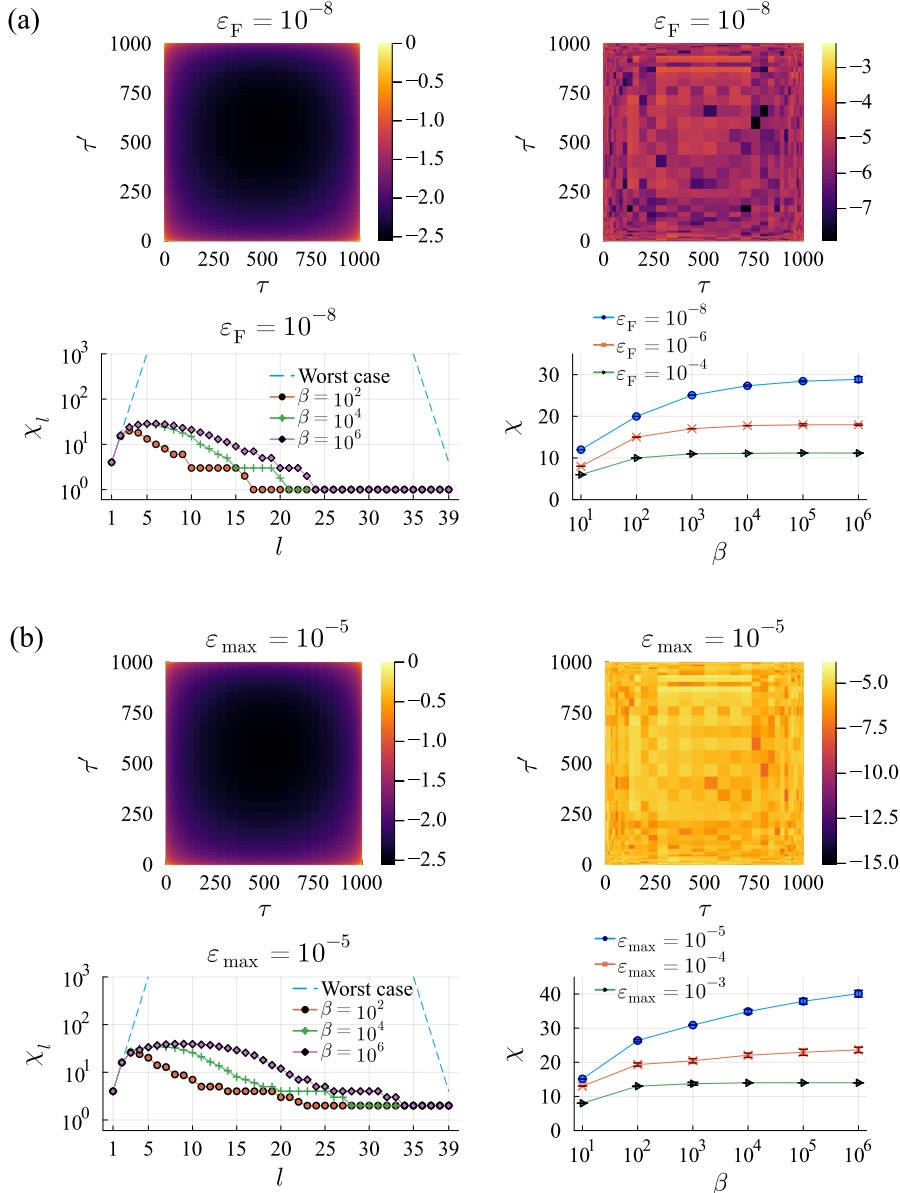

Figure 5: Results for $G^{\mathrm{FF}}(\tau, \tau')$: (a) with $\epsilon_{\mathrm{F}}$, (b) with $\epsilon_{\max}$. The colormaps show the $\log_{10}$ of the absolute value of the original (reference) data (left) and the absolute error (right) for a typical sample at $\beta = 10^3$ and $\mathcal{R} = 40$ [the same sample is shown in (a) and (b)]. In the colormaps, the data are normalized by the absolute maximum of the original data. The bottom two panels show sample-averaged bond dimensions $\chi_l$ and $\chi$, with 30 samples. The broken lines in the middle panels denote the worst exponential growth of $\chi_l$ for incompressible data.

## 5 Summary

In this paper, we have investigated the compactness of the imaginary-time Green's function in the QTT framework. The QTT representations of the Green's functions generated by random pole models were constructed by SVD and TCI with a desired cutoff in the Frobenius norm or the maximum norm, respectively.

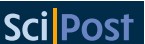

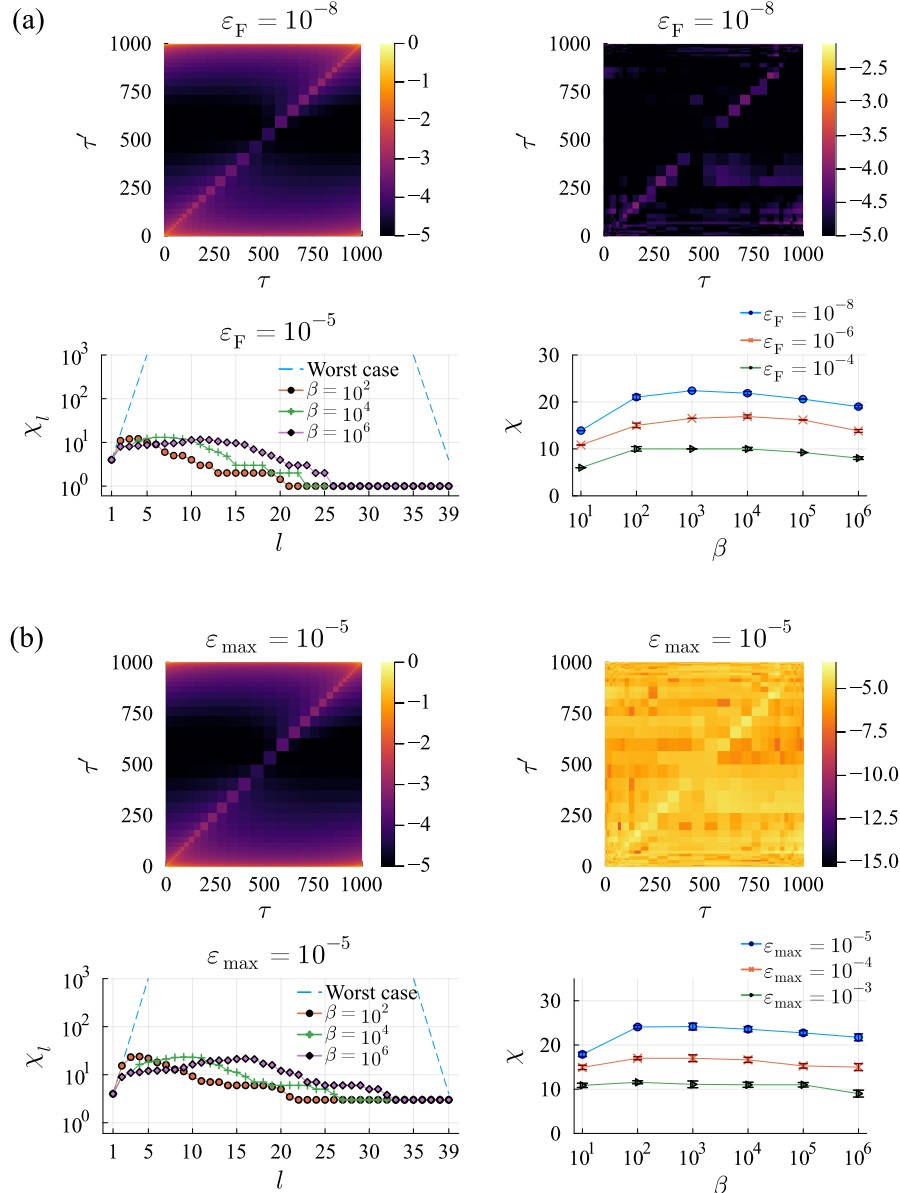

Figure 6: Results for $G^{\mathrm{FB}}(\tau, \tau')$: (a) with $\epsilon_{\mathrm{F}}$, (b) with $\epsilon_{\mathrm{max}}$. The colormaps show the $\log_{10}$ of the absolute value of the original (reference) data (left) and the absolute error (right) for a typical sample In the colormaps, the data are normalized by the absolute maximum of the original data. at $\beta = 10^3$ and $\mathcal{R} = 40$ [the same sample is shown in (a) and (b)]. There is a discontinuity line at $\tau = \tau'$. The bottom two panels show sample-averaged bond dimensions $\chi_l$ and $\chi$, with 30 samples. The broken lines in the middle panels denote the worst exponential growth of $\chi_l$ for incompressible data.

For one-time/-frequency objects, we mainly considered fermionic cases. A remarkable finding is that the bond dimension $\chi$ of the Green's function saturates at large $\beta$. We provided a counting-number argument to explain the saturation. A more rigorous explanation based on the multiscale interpolative construction of QTTs [31] is desired. We further found that a one-time object's bond dimension $\chi$ is smaller than that of the corresponding one-frequency object at small $\beta$ truncated in the Frobenius norm. This indicates that the imaginary-time representation is more favorable regarding data size at low temperatures.

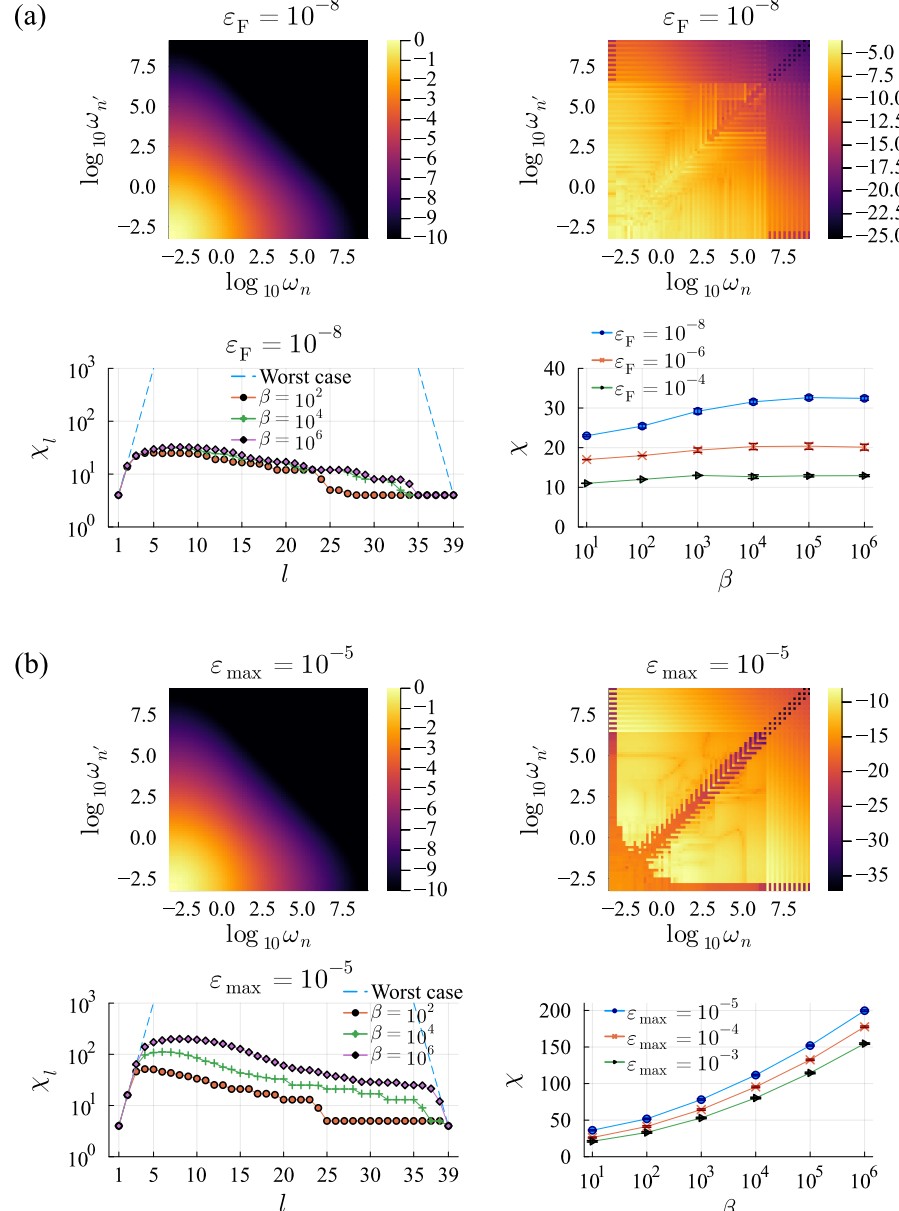

Figure 7: Results for $G^{\mathrm{FF}}(i\omega_n, i\omega_{n'})$: (a) with $\epsilon_{\mathrm{F}}$, (b) with $\epsilon_{\mathrm{max}}$. The colormaps show the $\log_{10}$ of the absolute value of the original (reference) data (left) and the absolute error (right) for a typical sample at $\beta = 10^3$ and $\mathcal{R} = 40$ [the same sample is shown in (a) and (b)]. In the colormaps, the data are normalized by the absolute maximum of the original data. Only the data for $\omega_n > 0$ and $\omega_{n'} > 0$ are shown. The bottom two panels show sample-averaged bond dimensions $\chi_l$ and $\chi$, with 30 samples. The broken lines in the middle panels denote the worst exponential growth of $\chi_l$ for incompressible data.

Furthermore, we analyzed two-time/-frequency objects generated by two distinct random pole models: the fermion-fermion and fermion-boson models. We examined the bond dimension for both models and truncation schemes for the imaginary-time and imaginary-frequency domains. The bond dimension $\chi$ grows rapidly at large $\beta$ for the imaginary-frequency domain with the truncation in terms of the maximum norm. This can be attributed to the divergence of the Green's function at zero frequency toward large $\beta$. In all other cases, $\chi$ mildly depends

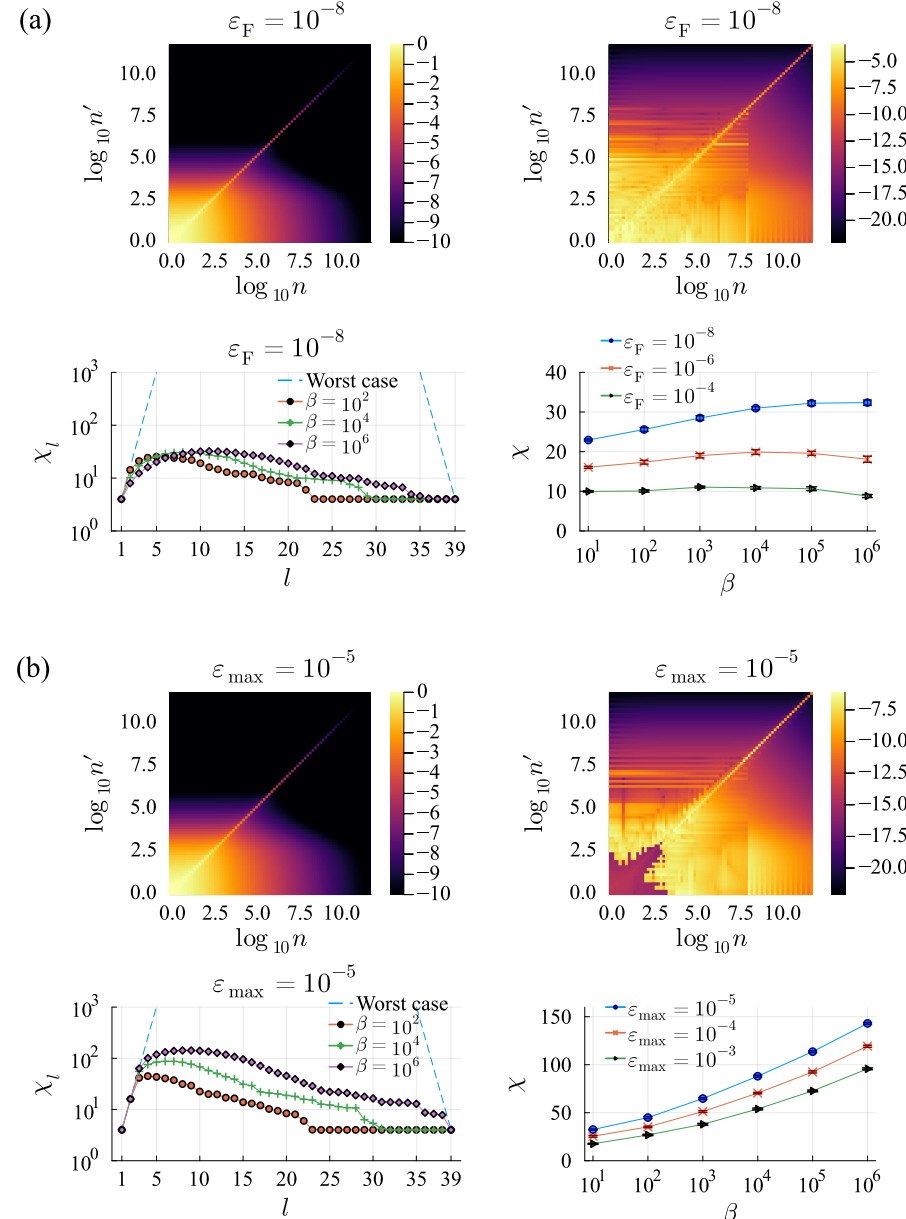

Figure 8: Results for $G^{\mathrm{FB}}(\mathrm{i}\omega_n, \mathrm{i}\omega_{n'})$: (a) with $\epsilon_{\mathrm{F}}$, (b) with $\epsilon_{\mathrm{max}}$. The colormaps show the $\log_{10}$ of the absolute value of the original (reference) data (left) and the absolute error (right) for a typical sample at $\beta = 10^3$ and $\mathcal{R} = 40$ [the same sample is shown in (a) and (b)]. In the colormaps, the data are normalized by the absolute maximum of the original data. Only the data for $\omega_n > 0$ and $\omega_{n'} > 0$ are shown. The bottom two panels show sample-averaged bond dimensions $\chi_l$ and $\chi$, with a total of 30 samples. The broken lines in the middle panels denote the worst exponential growth of $\chi_l$ for incompressible data.

on $\beta$, and even slightly decreases at large $\beta$. The origin of this unexpected behavior remains to be solved. The QTT representations of the fermion-boson model require smaller bond dimensions than those for the fermion-fermion model, which is consistent with the fact that a bosonic propagator can contain less information than a fermionic one.

Before closing, we would like to mention that the present study is a first step toward understanding the compactness of the imaginary-time Green's function in the QTT framework. The present study is limited to the Green's functions without momentum dependence. In the presence of momentum dependence, the computational complexity of the Green's function in the QTT framework is expected to be higher. A previous numerical study [29] indicates that the bond dimension $\chi$ grows roughly as $O(\sqrt{\beta})$ when the QTT representation is used to describe the two-dimensional momentum dependence of the one-particle Green's function. Further investigations are required to understand the computational complexity of the Green's function with both of frequency and momentum dependence in the QTT framework.

Many-body calculation at the two-particle level [35–40] requires the treatment of three-point/four-point correlation and vertex functions, which could contain terms leading to divergence of the Frobenius norm in the large-$\mathcal{R}$ limit ($\mathcal{R}$ is the number of bits). Examples include a constant term in the imaginary-frequency domain. We may have to treat such terms separately or resort to truncation in the maximum norm. In particular, solving the Bethe-Salpeter equation requires matrix multiplications with two-particle objects. A recently proposed TCI algorithms [27] may be a promising approach to perform matrix multiplications using truncation in the maximum norm. A performance comparison with the conventional methods for handling two-particle objects, such as the overcomplete IR [12, 15] and DLR [41], is also desired.

In the present study, we have not considered the effect of index permutations in the QTT representation. A combination of QTT and automatic structure optimization of tensor networks [42] may be an interesting future direction.

## Acknowledgments

We used `SparseIR.jl` [43] for generating sparse grids, `TensorCrossInterpolation.jl` for TCI [27], and `ITensors.jl` [44]. The authors are grateful to Anna Kauch for critically reading the manuscript. H.S. thanks Jason Kaye, Marc. K. Ritter, Makus Wallerberger, and Michael Lindsey for fruitful discussions on multiscale interpolative construction of QTTs.

**Funding information**   H.S. was supported by JSPS KAKENHI Grants No.18H01158, No. 21H01041, No. 21H01003, and No. 23H03817 as well as JST PRESTO Grant No. JP-MJPR2012, Japan.

## A   Results of one-time/-frequency objects for different noise models

We show results for a different noise model, where $c_p$ is taken from a uniform distribution on $[-1/2, 1/2]$. The coefficients $c_p$ are normalized such that $\sum_p c_p = 1$. We obtained the same results qualitatively by using this noise model. Figure 9 shows typical results for fermionic $G^{\mathrm{F}}(\tau)$ obtained with $\epsilon_{\mathrm{max}} = 10^{-5}$. The $\beta$ dependence of $\chi$ is qualitatively the same as those obtained for non-negative $c_p$. A small dip around $\tau = 1000$ originates from a sign change in $G^{\mathrm{F}}(\tau)$.

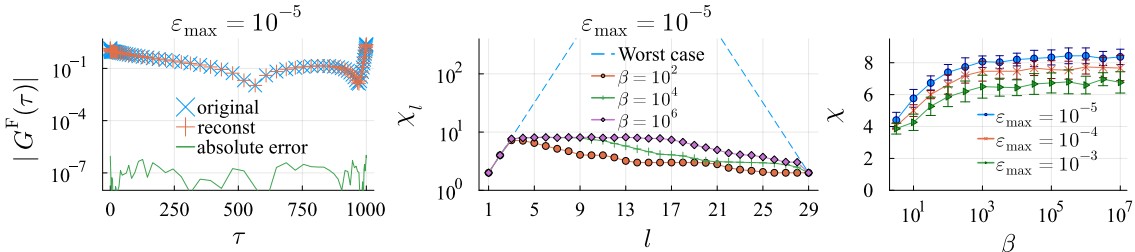

Figure 9: Results for fermionic $G^{\mathrm{F}}(\tau)$ obtained by the truncation using $\epsilon_{\mathrm{max}}$, where $c_p$ can be positive or negative.

## B Results for bosonic $G^{\mathrm{B}}(\tau)$ and $G^{\mathrm{B}}(i\omega)$

Figure 10 shows results obtained for bosonic $G^{\mathrm{B}}(\tau)$ and $G^{\mathrm{B}}(i\omega)$ with $\epsilon_{\mathrm{max}} = 10^{-5}$. We used the same pole model but dropped the pole at $\omega = 0$. Here, we show only results obtained with truncation in terms of the maximum norm as we obtained similar results with the two truncation schemes. One can see that $\chi_l$ and $\chi$ show behaviors similar to the fermionic cases.

## C Results of two-time/-frequency objects for different noise models

We show results for a different noise model in Fig. 11, where $c_{pp'}$ is taken from a uniform distribution on $[-1/2, 1/2]$. The coefficients $c_{pp'}$ are normalized such that $\sum_{pp'} c_{pp'} = 1$. The overall behavior is similar to that obtained for non-negative $c_{pp'}$ shown in Fig. 5. For the truncation in terms of the maximum norm, the bond dimension $\chi$ grows more rapidly than the non-negative $c_{pp'}$ case. This is because the maximum norm of the Green's function is larger in the positive-negative $c_{pp'}$ case, and the same $\epsilon_{\mathrm{max}}$ becomes more strict effectively.

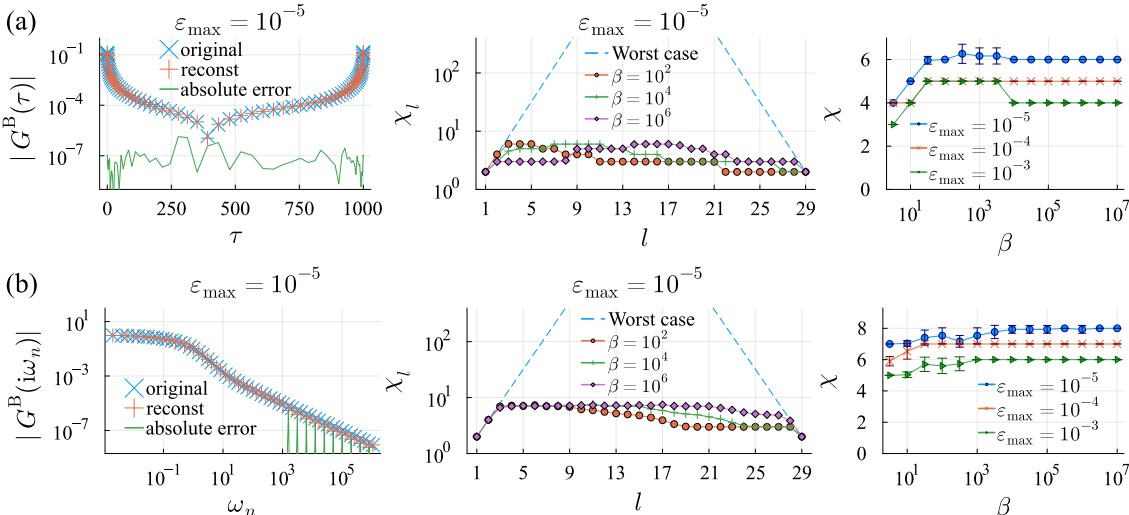

Figure 10: Results for bosonic Green's function obtained by the truncation using $\epsilon_{\mathrm{max}}$: (a) $G^{\mathrm{B}}(\tau)$, and (b) $G^{\mathrm{B}}(i\omega)$.

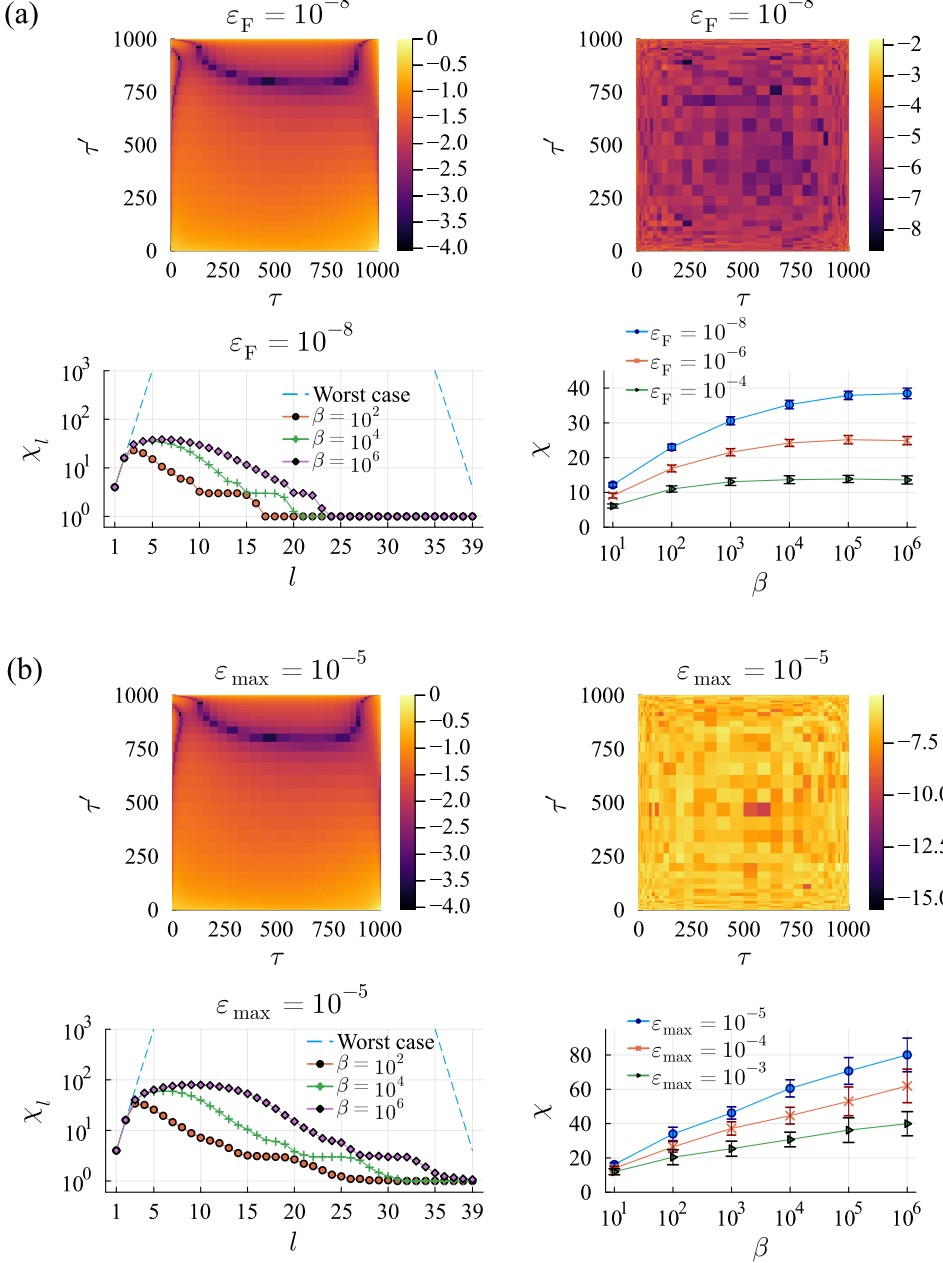

Figure 11: Results for fermionic $G^{\mathrm{FF}}(\tau, \tau')$: (a) with $\epsilon_{\mathrm{F}}$, (b) with $\epsilon_{\mathrm{max}}$. where $c_p$ can be positive or negative. The colormaps show the $\log_{10}$ of the absolute value of the original (reference) data (left) and the absolute error (right) for a typical sample at $\beta = 10^3$ and $\mathcal{R} = 40$ [the same sample is shown in (a) and (b)]. In the colormaps, the data are normalized by the absolute maximum of the original data. The bottom two panels show sample-averaged bond dimensions $\chi_l$ and $\chi$, with 30 samples. The broken lines in the middle panels denote the worst exponential growth of $\chi_l$ for incompressible data.

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
