# Peer review of "Compactness of quantics tensor train representations of local imaginary-time propagators"

_SciPost Physics, doi:SciPost Phys. 18, 007 (2025)_

## Round 1 · Referee Report · Anonymous (Referee 1) · 2024-4-8

Report
The authors investigate the quantics tensor train (QTT) representation of local imaginary-time and -frequency propagators. They consider one- and two-time/frequency objects. They find that - in certain cases - the considered objects are compressible with a bond dimension that remains finite even for inverse temperature going to infinity.
At the present stage I do not believe that the paper has enough substance to warrant publication in SciPost Physics. There are two main reasons:
(1) The paper remains on a superficial level. Whenever an interesting observation is made a thorough analysis thereof is delegated to future work. Particularly for two-frequency objects, Figs. 7 and 8, it seems that the Frobenius-norm approach has small bond dimension but large errors (the color plot for the absolute error looks similar to that of the absolute values) while the maximum-norm approach gives small errors in the color plots but has bond dimensions growing indefinitely. The question whether two-frequency objects are QTT compressible or not - crucial for the scope of the paper - remains unclear.
(2) Generally it seems insufficient to me to consider random pole models to judge the compressibility of many-body functions. Particularly in the field of analytic continuation it is a common problem that new methods are presented with excellent results on pole models and later turn out not remotely as successful for models encountered in 'real life'. Therefore it is advisable to study realistic models too. Here these could be obtained e.g. in dynamical mean-field theory with quantum Monte Carlo solvers. Moreover one should also consider non Fermi liquid scenarios, possibly using simple but typical analytic expression.
More detailed criticism is as follows:
- In the first paragraph of 3.2 it is unclear how the grid of Matsubara frequencies is mapped to the bit representation. Which values does n take? How is the grid from -infinity to infinity mapped to an interval from 0 to 1?
- In Sec. 4 I do not understand the motivation for analyzing Eq. (12) and Eq. (14) separately. According to the Lehmann representation a generic two-frequency propagator is a superposition of Eq. (12) and Eq. (14). Moreover the authors did not address the presence of anomalous terms proportional to delta_{iOmega,0} where Omega is a bosonic Matsubara frequency.
- In the last paragraph of Sec. 4.3.2 the authors state that the two-frequency object G^FB has less information than G^FF but their explanation is based on one-frequency objects. It is unclear why properties of the latter carry over to the former.
- The color plots in Figs. 5-8 are not well formatted. It is unclear how to extract from these plots whether the QTT representation works or does not work. For instance in Fig. 6a the absolute values seem to have the same size as the absolute errors since both color bars go up to -2 (or even -1).
- In the last paragraph of Sec. 5 the authors mention the presence of constant terms in vertex functions. They state that the single-/multi-boson exchange (SBE/MBE) framework does not involve constant terms. I cannot follow here for two reasons: First the SBE/MBE framework does contain constant terms, namely the bosonic propagator contains the bare interaction and the Hedin vertex contains a term that equals unity. Second why is the SBE/MBE framework singled out here? There are many other frameworks that similarly have constant terms which similarly can be subtracted.
Author: Hiroshi Shinaoka on 2024-04-13 [id 4416]
(in reply to Report 1 on 2024-04-08)
We greatly appreciate your efforts on reviewing the manuscript. Your comments are very useful to improve the readability of the manuscript, although we disagree with the conclusions that the paper has enough substance to warrant publication in SciPost Physics.
We are preparing a revised manuscript, but the SciPost platform allows the authors to communicate with a referee online interactively. Please let us address the main criticism before an extensive revision of the manuscript.
(1) The paper remains on a superficial level. Whenever an interesting observation is made a thorough analysis thereof is delegated to future work. Particularly for two-frequency objects, Figs. 7 and 8, it seems > that the Frobenius-norm approach has small bond dimension but large errors (the color plot for the absolute error looks similar to that of the absolute values) while the maximum-norm approach gives small errors in the color plots but bond dimensions growing indefinitely. The question whether two-frequency objects are QTT compressible or not - crucial for the scope of the paper - remains unclear.
The scaling plots in Fig. 6-8 strongly indicate that the two-frequency objects are compressible. We agree that the colormaps are to be improved. We attached a revised version of Fig. 6 for a smaller cutoff of 1e-8 in this reply. This cutoff corresponds to 4-digit accuracy [Eq. (8)]. The new figure demonstrates the compressibility of the two-time object more clearly.
We believe that our non-trivial numerical finding will stimulate further (mathematical) studies. The bond dimensions remain constant at low T in the Frobenius norm case, leading to the data size scaling O(chi^2 log beta) = O(log beta). The results surprisingly indicate that the QTT approach superpasses the scaling of the conventional approach (i.e.,IR, O((log beta)^2).
The max-norm case is very singular; since the function is diverging, we should avoid using the max norm. Since TCI has been attracting more attention, this result is useful to other many-body theorists as a red flag.
(2) Generally it seems insufficient to me to consider random pole models to judge the compressibility of many-body functions. Particularly in the field of analytic continuation it is a common problem that new methods are presented with excellent results on pole models and later turn out not remotely as successful for models encountered in 'real life'. Therefore it is advisable to study realistic models too. Here these could be obtained e.g. in dynamical mean-field theory with quantum Monte Carlo solvers. Moreover one should also consider non Fermi liquid scenarios, possibly using simple but typical analytic expression.
Thank you for raising this important point. We should have explained the motivation of using the random-pole model in more detail.
We used the random-pole model to generate random imaginary-time data. In our analysis, we deal with only the imaginary-time data but never reconstruct the spectral functions. If the pole positions are chosen appropriately, we could reproduce an imaginary-time function generated by an any continuous spectral function precisely, according to previous studies on IR and DLR. This setup is clearly different from the numerical analytic continuation, where whether or not the original spectral function is discrete or continuous can make significant differences in the quality of the reconstruction of the spectral function.
Our random-pole model is more challenging than “real-life” data. We have already done extensive numerical analysis of real-life data from various numerical simulations such as, real-frequency spectral functions of Kondo impurity models, vertex functions from DFT+DMFT, nonequilibrium DMFT and more in PRX 13, 021015 (2023). However, analyzing data from numerical simulations is not enough to understand the intrinsic properties of the QTT representation. For example, QMC data include statistical noise, hindering analysis at low temperature. The Kondo spectral function is too simple and contains only a few peaks (Fig. 13 of the PRX). In contrast, our random pole model contains many poles, whose number increases at low temperature. This allows us to encode more and more information into the random pole model through many random coefficients. We will update the manuscript to stress these points, i.e., our test cases are much more challenging to QTT than real-life cases.
Attachment:
two_svd_tau_rotated_cut864_8_5.pdf
Anonymous on 2024-09-22 [id 4796]
(in reply to Hiroshi Shinaoka on 2024-04-13 [id 4416])
The authors investigate the quantics tensor train (QTT) representation of local imaginary-time and -frequency propagators. They > consider one- and two-time/frequency objects. They find that - in certain cases - the considered objects are compressible with a bond > dimension that remains finite even for inverse temperature going to infinity.
We thank Referee A for recognizing the most importance numerical finding in the present study.
At the present stage I do not believe that the paper has enough substance to warrant publication in SciPost Physics. There are two main reasons:
(1) The paper remains on a superficial level. Whenever an interesting observation is made a thorough analysis thereof is delegated to future work. Particularly for two-frequency objects, Figs. 7 and 8, it seems that the Frobenius-norm approach has small bond dimension but large errors (the color plot for the absolute error looks similar to that of the absolute values) while the maximum-norm approach gives small errors in the color plots but has bond dimensions growing indefinitely. The question whether two-frequency objects are QTT compressible or not - crucial for the scope of the paper - remains unclear.
We thank Referee A for the comments.
We want to address the comment regarding the superficial level of the paper. Our numerical experiments indicate that multi-frequency/-time objects are compressible in QTT. Moreover, the QTT approach outperforms current state-of-the-art representations, such as the intermediate representation and the discrete Lehman representation for two-time/-frequency cases asymptotically.
Since the submission of the first manuscript, we have been informed of a recent applied mathematics preprint entitled “Multiscale interpolative construction of quantized tensor trains” (arXiv:2311.12554). After discussions with the author of this preprint, we recognized that this new mathematical approach may provide insights into why the bond dimension saturates for “one-frequency/-time objects.” However, this framework does not explain the numerical results for multi-frequency/-time objects. This indicates that our numerical findings are highly nontrivial, and providing a comprehensive explanation is beyond the scope of the current study.
Considering these circumstances, we believe it is scientifically fair to present the numerical evidence at this point and offer all possible arguments.
We admit that the compressibility was not clearly visible in some of the color maps in the original manuscript.
In response to the comment, we have improved the manuscript in the following ways:
- Revised the abstract to emphasize the main findings of the present work.
- Cited the applied mathematics paper (arXiv:2311.12554) and discussions.
- Improved visibility of colormaps in Figs. 5, 6, 7, 8 with smaller cutoffs and appropriate color ranges.
We hope these revisions address the concerns raised by Referee A and clarify the contributions of our study.
(2) Generally it seems insufficient to me to consider random pole models to judge the compressibility of many-body functions. Particularly in the field of analytic continuation it is a common problem that new methods are presented with excellent results on pole models and later turn out not remotely as successful for models encountered in 'real life'. Therefore it is advisable to study realistic models too. Here these could be obtained e.g. in dynamical mean-field theory with quantum Monte Carlo solvers. Moreover one should also consider non Fermi liquid scenarios, possibly using simple but typical analytic expression.
We thank Referee A for highlighting this important point.
We would like to address the comment regarding the sufficiency of random pole models for judging the compressibility of many-body functions. While we understand the concern, we respectfully disagree with the criticism that random pole models are inadequate for this purpose.
Numerical analytic continuation is indeed an ill-conditioned process, and its success is highly model-dependent. However, there is no direct correlation between numerical analytic continuation and the compression of the Green’s function. In terms of compression, numerous “real-life” data have been analyzed in our previous study (PRX 13, 021015 (2023)), including spectral functions of Anderson impurity models (Fig. 13) and vertex functions of the Hubbard atom (Fig. 14). These real-life data typically have simpler features and are expected to be more compressible than random pole models.
The primary objective of the present study is to investigate worst-case scenarios for compression using randomly generated Green’s functions. Our model Green’s function contains increasing amounts of information as the number of poles grows logarithmically with respect to inverse temperature, and the coefficients are random. This random model analysis also allowed us to explore extremely low temperatures, which was not possible in the previous study. Our numerical simulations demonstrated that even these hardly compressible Green’s functions are highly compressible in QTT.
In response to the comment, we have added the following sentences to the abstract to emphasize that random-pole models are appropriate for worst-case scenarios:
``To study worst-case scenarios, we employ random pole models, where the number of poles grows logarithmically with the inverse temperature and coefficients are random. The Green's functions generated by these models are expected to be more difficult to compress than those from physical systems. The numerical analysis reveals that these propagators are highly compressible in QTT, outperforming the state-of-the-art approaches such as intermediate representation and discrete Lehmann representation.''
- In the first paragraph of 3.2 it is unclear how the grid of Matsubara frequencies is mapped to the bit representation. Which values does n take? How is the grid from -infinity to infinity mapped to an interval from 0 to 1?
We thank Referee A for pointing out an important point. The integer $n$ takes $-2^{R-1}, -2^{R-1}+1, \cdots, 2^{R-1} - 2, 2^{R-1} - 1$. We have added the following sentence in Section 3.2:
``The values of $n+2^{R-1}$ run from 0 to $2^R-1$.''
- In Sec. 4 I do not understand the motivation for analyzing Eq. (12) and Eq. (14) separately. According to the Lehmann representation a generic two-frequency propagator is a superposition of Eq. (12) and Eq. (14). Moreover the authors did not address the presence of anomalous terms proportional to delta_{iOmega,0} where Omega is a bosonic Matsubara frequency.
We thank Referee A for pointing out important points. In QTT, a superposition of Eq. (12) and Eq. (14) simply add up bond dimensions for these contribution, which does not change the scaling with respect to temperature. The contribution from anomalous terms proportional can be represented as a TT of bond dimension 1, which increases the bond dimension of a total bosonic propagator only by 1. We agree that these points are confusing for the readers who are not familiar with QTTs. We have added the following sentences in Section 4.1:
``We analyze these two models separately because their superposition only sums up the bond dimensions of the two models (see Sec. 2.3), not affecting scaling behaviors with $\beta$. We do not consider a constant term in the Matsubara frequency space and anomalous propagator terms at zero bosonic frequency because these contributions can be represented by a single TT of bond dimension 1.''
- In the last paragraph of Sec. 4.3.2 the authors state that the two-frequency object G\^FB has less information than G\^FF but their explanation is based on one-frequency objects. It is unclear why properties of the latter carry over to the former. We thank Referee A for pointing out the confusing statement.
We have added the following sentences in the last paragraph of Sec. 4.3.2 to connect these two points.
``Although the above argument is based on single-frequency objects, it is expected to hold for two-frequency objects as well. To be more specific, two-frequency objects have a similar structure concerning bosonic frequency dependence [see $w_{p’}$ in the numerator of Eq.~(14)]. This may explain the smaller bond dimension of $G^\mathrm{FB}(\tau, \tau’)$ compared to $G^\mathrm{FF}(\tau, \tau’)$.''
- The color plots in Figs. 5-8 are not well formatted. It is unclear how to extract from these plots whether the QTT representation works or does not work. For instance in Fig. 6a the absolute values seem to have the same size as the absolute errors since both color bars go up to -2 (or even -1).
We thank Referee A for pointing out the readability issues. We have replaced Fig. 5-8 with new colormaps with smaller cutoffs and color ranges to demonstrate the compressibility of the QTT reprensetation more clearly.
- In the last paragraph of Sec. 5 the authors mention the presence of constant terms in vertex functions. They state that the single-/multi-boson exchange (SBE/MBE) framework does not involve constant terms. I cannot follow here for two reasons: First the SBE/MBE framework does contain constant terms, namely the bosonic propagator contains the bare interaction and the Hedin vertex contains a term that equals unity. Second why is the SBE/MBE framework singled out here? There are many other frameworks that similarly have constant terms which similarly can be subtracted.
We thank Referee A for pointing out the unclear statement. We did not intend to single out the SBE/MBE framework. What is important is that we may have to subtract the constant terms from the vertex functions in calculations at the two-particle level. In the revised manuscript, we cite more references on calculations at the two-particle level and clarify the necessity of subtracting constant terms.
Author: Hiroshi Shinaoka on 2024-09-22 [id 4797]
(in reply to Report 2 on 2024-04-24)We thank Referee B for accepting the possible broad interest of our manuscript. We believe that our result will possibly attract attention even from the applied math community. To increase the visibility, we have cited a recent math paper ``Multiscale interpolative construction of quantized tensor trains'' (Michael Lindsey, arXiv:2311.12554).
We thank Referee B for the suggestion. Although our random-pole model has a discrete spectral function, the discrete Lehmann representation can accurately span a Green's function generated by a continuous spectral function. In addition, our model has random coefficients, and thus is expected to be more difficult to compress by QTT than physical Green's functions, which are usually have regular structures.
This comment is related to the criticism from Referee A:"random-pole model may not suffice to judge the compressibility of many-body functions.'' To state the applicability of the present numerical results, we have added the following sentences to the abstract.
"To study worst-case scenarios, we employ random pole models, where the number of poles grows logarithmically with the inverse temperature and coefficients are random. The Green's functions generated by these models are expected to be more difficult to compress than those from physical systems. The numerical analysis reveals that these propagators are highly compressible in QTT, outperforming the state-of-the-art approaches such as intermediate representation and discrete Lehmann reprensentation.''
We thank Referee B for the suggestion. We have cited extensive works based on QTTs outside condensed matter physics in the introduction, e.g., turbulence, plasma physics.
We thank Referee B for asking an interesting and important question. To the best of our knowledge, noone has investigated the index-ordering dependence of the QTT representation for imaginary-time propagators. Our simple choice for the ordering leads to significantly compact representations. We think that the optimal ordering may depend on the system. A recent paper proposed an automated algorithm for optimizing the topology and index ordering for tensor networks [T. Hikihara \textit{et al.}, PRR 5, 013031 (2023)]. Combining quantics and the structure optimization algorithm will be a fascinating feature study. We have added a short discussion on this point in the summary section.
"In the present study, we have not considered the effect of index permutations in the QTT representation. A combination of QTT and automatic structure optimization of tensor networks~[42] may be an interesting future direction.''
We thank Referee B for pointing out the insufficient description. Computing dynamical responses of correlated systems has been a challenging issue in condensed matter physics. This requires efficiently solving diagrammatic equations at the two-particle level, including the Bethe-Salpeter equation. In addition, solving the Bethe-Salpeter equation requires matrix multiplications of two-particle objects, which is more challenging than compressing objects. We have revised and extended the second last paragraph in the summary section.
We thank Referee B for the suggestion. The dip is due to a sign change in $G(\tau)$, i.e., the cancellation of contributions from positive and negative coefficients. We have added an explanation where Fig. 9 is mentioned (Appendix A).
"A small dip around $\tau=1000$ originates from a sign change in $G^\mathrm{F}(\tau)$.''

---

## Round 1 · Referee Report · Anonymous (Referee 2) · 2024-4-24

Strengths
1. Of possible broad interest
Weaknesses
1. Requires additional details about range of applicability and physical rationale
Report
The manuscript provides valuable insight into the compactness of local imaginary-time propagators in quantics tensor train format. The work is written with the quantum field theory community in mind, but given its focus on tensor trains could have relatively broad readership in mathematics, physics, chemistry, etc. with changes. It would also improve the paper to clarify not only the findings, but the key breakthrough they wish to highlight (with regards to literature in SVD/TCI, etc.).
Requested changes
1. It would be helpful to provide more information (and physical intuition) to explain why the numerical results for the models expected here are expected to be more broadly applicable.
2. The paper could have a broader impact by further referencing the extensive work being done outside of quantum field theory with imaginary-time propagators. Connection to MPS would also help.
3. Can a statement be made about how the reshaping order is chosen in the general case (i.e., in applications where there are concerns that QTT rank depends significantly on ordering)? Are there additional calculations to support whether reordering significantly affects the presented results?
4. It would be beneficial to provide a rationale for why the Bethe-Salpeter equation is an important future direction.
5. When Figure 9 is mentioned, it would help to provide an explanation for the dip near 1000, which is distinct from all other models presented.
Recommendation
Ask for minor revision

---

## Round 2 · Referee Report · Anonymous (Referee 1) · 2024-11-11

Report

The authors have addressed all my concerns and the paper can be published as is.

Recommendation

Publish (meets expectations and criteria for this Journal)

---

## Round 2 · Author Response

Dear Editor,

Thank you for sending us the referee reports for our manuscript -- Compactness of quantics tensor train representations of local imaginary-time propagators.

We thank the referees for the careful reading of the manuscript and the useful feedback, which we have taken into account in the resubmitted manuscript. We address below the specific suggestions and criticism, and hope that with these clarifications and the improvements in the manuscript, our work can be accepted for publication in SciPost.

Response to Referee A

The authors investigate the quantics tensor train (QTT) representation of local imaginary-time and -frequency propagators. They > consider one- and two-time/frequency objects. They find that - in certain cases - the considered objects are compressible with a bond > dimension that remains finite even for inverse temperature going to infinity.

We thank Referee A for recognizing the most importance numerical finding in the present study.

At the present stage I do not believe that the paper has enough substance to warrant publication in SciPost Physics. There are two main reasons:

(1) The paper remains on a superficial level. Whenever an interesting observation is made a thorough analysis thereof is delegated to future work. Particularly for two-frequency objects, Figs. 7 and 8, it seems that the Frobenius-norm approach has small bond dimension but large errors (the color plot for the absolute error looks similar to that of the absolute values) while the maximum-norm approach gives small errors in the color plots but has bond dimensions growing indefinitely. The question whether two-frequency objects are QTT compressible or not - crucial for the scope of the paper - remains unclear.

We thank Referee A for the comments.

We want to address the comment regarding the superficial level of the paper. Our numerical experiments indicate that multi-frequency/-time objects are compressible in QTT. Moreover, the QTT approach outperforms current state-of-the-art representations, such as the intermediate representation and the discrete Lehman representation for two-time/-frequency cases asymptotically.

Since the submission of the first manuscript, we have been informed of a recent applied mathematics preprint entitled “Multiscale interpolative construction of quantized tensor trains” (arXiv:2311.12554). After discussions with the author of this preprint, we recognized that this new mathematical approach may provide insights into why the bond dimension saturates for “one-frequency/-time objects.” However, this framework does not explain the numerical results for multi-frequency/-time objects. This indicates that our numerical findings are highly nontrivial, and providing a comprehensive explanation is beyond the scope of the current study.

Considering these circumstances, we believe it is scientifically fair to present the numerical evidence at this point and offer all possible arguments.

We admit that the compressibility was not clearly visible in some of the color maps in the original manuscript.

In response to the comment, we have improved the manuscript in the following ways:

  • Revised the abstract to emphasize the main findings of the present work.
  • Cited the applied mathematics paper (arXiv:2311.12554) and discussions.
  • Improved visibility of colormaps in Figs. 5, 6, 7, 8 with smaller cutoffs and appropriate color ranges.

We hope these revisions address the concerns raised by Referee A and clarify the contributions of our study.

(2) Generally it seems insufficient to me to consider random pole models to judge the compressibility of many-body functions. Particularly in the field of analytic continuation it is a common problem that new methods are presented with excellent results on pole models and later turn out not remotely as successful for models encountered in 'real life'. Therefore it is advisable to study realistic models too. Here these could be obtained e.g. in dynamical mean-field theory with quantum Monte Carlo solvers. Moreover one should also consider non Fermi liquid scenarios, possibly using simple but typical analytic expression.

We thank Referee A for highlighting this important point.

We would like to address the comment regarding the sufficiency of random pole models for judging the compressibility of many-body functions. While we understand the concern, we respectfully disagree with the criticism that random pole models are inadequate for this purpose.

Numerical analytic continuation is indeed an ill-conditioned process, and its success is highly model-dependent. However, there is no direct correlation between numerical analytic continuation and the compression of the Green’s function. In terms of compression, numerous “real-life” data have been analyzed in our previous study (PRX 13, 021015 (2023)), including spectral functions of Anderson impurity models (Fig. 13) and vertex functions of the Hubbard atom (Fig. 14). These real-life data typically have simpler features and are expected to be more compressible than random pole models.

The primary objective of the present study is to investigate worst-case scenarios for compression using randomly generated Green’s functions. Our model Green’s function contains increasing amounts of information as the number of poles grows logarithmically with respect to inverse temperature, and the coefficients are random. This random model analysis also allowed us to explore extremely low temperatures, which was not possible in the previous study. Our numerical simulations demonstrated that even these hardly compressible Green’s functions are highly compressible in QTT.

In response to the comment, we have added the following sentences to the abstract to emphasize that random-pole models are appropriate for worst-case scenarios:

``To study worst-case scenarios, we employ random pole models, where the number of poles grows logarithmically with the inverse temperature and coefficients are random. The Green's functions generated by these models are expected to be more difficult to compress than those from physical systems. The numerical analysis reveals that these propagators are highly compressible in QTT, outperforming the state-of-the-art approaches such as intermediate representation and discrete Lehmann representation.''

  • In the first paragraph of 3.2 it is unclear how the grid of Matsubara frequencies is mapped to the bit representation. Which values does n take? How is the grid from -infinity to infinity mapped to an interval from 0 to 1?

We thank Referee A for pointing out an important point. The integer $n$ takes $-2^{R-1}, -2^{R-1}+1, \cdots, 2^{R-1} - 2, 2^{R-1} - 1$. We have added the following sentence in Section 3.2:

``The values of $n+2^{R-1}$ run from 0 to $2^R-1$.''

  • In Sec. 4 I do not understand the motivation for analyzing Eq. (12) and Eq. (14) separately. According to the Lehmann representation a generic two-frequency propagator is a superposition of Eq. (12) and Eq. (14). Moreover the authors did not address the presence of anomalous terms proportional to delta_{iOmega,0} where Omega is a bosonic Matsubara frequency.

We thank Referee A for pointing out important points. In QTT, a superposition of Eq. (12) and Eq. (14) simply add up bond dimensions for these contribution, which does not change the scaling with respect to temperature. The contribution from anomalous terms proportional can be represented as a TT of bond dimension 1, which increases the bond dimension of a total bosonic propagator only by 1. We agree that these points are confusing for the readers who are not familiar with QTTs. We have added the following sentences in Section 4.1:

``We analyze these two models separately because their superposition only sums up the bond dimensions of the two models (see Sec. 2.3), not affecting scaling behaviors with $\beta$. We do not consider a constant term in the Matsubara frequency space and anomalous propagator terms at zero bosonic frequency because these contributions can be represented by a single TT of bond dimension 1.''

  • In the last paragraph of Sec. 4.3.2 the authors state that the two-frequency object G\^FB has less information than G\^FF but their explanation is based on one-frequency objects. It is unclear why properties of the latter carry over to the former. We thank Referee A for pointing out the confusing statement.

We have added the following sentences in the last paragraph of Sec. 4.3.2 to connect these two points.

``Although the above argument is based on single-frequency objects, it is expected to hold for two-frequency objects as well. To be more specific, two-frequency objects have a similar structure concerning bosonic frequency dependence [see $w_{p’}$ in the numerator of Eq.~(14)]. This may explain the smaller bond dimension of $G^\mathrm{FB}(\tau, \tau’)$ compared to $G^\mathrm{FF}(\tau, \tau’)$.''

  • The color plots in Figs. 5-8 are not well formatted. It is unclear how to extract from these plots whether the QTT representation works or does not work. For instance in Fig. 6a the absolute values seem to have the same size as the absolute errors since both color bars go up to -2 (or even -1).

We thank Referee A for pointing out the readability issues. We have replaced Fig. 5-8 with new colormaps with smaller cutoffs and color ranges to demonstrate the compressibility of the QTT reprensetation more clearly.

  • In the last paragraph of Sec. 5 the authors mention the presence of constant terms in vertex functions. They state that the single-/multi-boson exchange (SBE/MBE) framework does not involve constant terms. I cannot follow here for two reasons: First the SBE/MBE framework does contain constant terms, namely the bosonic propagator contains the bare interaction and the Hedin vertex contains a term that equals unity. Second why is the SBE/MBE framework singled out here? There are many other frameworks that similarly have constant terms which similarly can be subtracted.

We thank Referee A for pointing out the unclear statement. We did not intend to single out the SBE/MBE framework. What is important is that we may have to subtract the constant terms from the vertex functions in calculations at the two-particle level. In the revised manuscript, we cite more references on calculations at the two-particle level and clarify the necessity of subtracting constant terms.

Reply to Referee B

Strength: Of possible broad interest\ The manuscript provides valuable insight into the compactness of local imaginary-time propagators in quantics tensor train format. The work is written with the quantum field theory community in mind, but given its focus on tensor trains could have relatively broad readership in mathematics, physics, chemistry, etc. with changes. It would also improve the paper to clarify not only the findings, but the key breakthrough they wish to highlight (with regards to literature in SVD/TCI, etc.).

We thank Referee B for accepting the possible broad interest of our manuscript. We believe that our result will possibly attract attention even from the applied math community. To increase the visibility, we have cited a recent math paper ``Multiscale interpolative construction of quantized tensor trains'' (Michael Lindsey, arXiv:2311.12554).

  1. It would be helpful to provide more information (and physical intuition) to explain why the numerical results for the models expected here are expected to be more broadly applicable.

We thank Referee B for the suggestion. Although our random-pole model has a discrete spectral function, the discrete Lehmann representation can accurately span a Green's function generated by a continuous spectral function. In addition, our model has random coefficients, and thus is expected to be more difficult to compress by QTT than physical Green's functions, which are usually have regular structures.

This comment is related to the criticism from Referee A: ``random-pole model may not suffice to judge the compressibility of many-body functions.'' To state the applicability of the present numerical results, we have added the following sentences to the abstract.

``To study worst-case scenarios, we employ random pole models, where the number of poles grows logarithmically with the inverse temperature and coefficients are random. The Green's functions generated by these models are expected to be more difficult to compress than those from physical systems. The numerical analysis reveals that these propagators are highly compressible in QTT, outperforming the state-of-the-art approaches such as intermediate representation and discrete Lehmann reprensentation.''

  1. The paper could have a broader impact by further referencing the extensive work being done outside of quantum field theory with imaginary-time propagators. Connection to MPS would also help.

We thank Referee B for the suggestion. We have cited extensive works based on QTTs outside condensed matter physics in the introduction, e.g., turbulence, plasma physics.

  1. Can a statement be made about how the reshaping order is chosen in the general case (i.e., in applications where there are concerns that QTT rank depends significantly on ordering)? Are there additional calculations to support whether reordering significantly affects the presented results?

We thank Referee B for asking an interesting and important question. To the best of our knowledge, noone has investigated the index-ordering dependence of the QTT representation for imaginary-time propagators. Our simple choice for the ordering leads to significantly compact representations. We think that the optimal ordering may depend on the system. A recent paper proposed an automated algorithm for optimizing the topology and index ordering for tensor networks [T. Hikihara \textit{et al.}, PRR 5, 013031 (2023)]. Combining quantics and the structure optimization algorithm will be a fascinating feature study. We have added a short discussion on this point in the summary section.

``In the present study, we have not considered the effect of index permutations in the QTT representation. A combination of QTT and automatic structure optimization of tensor networks~[42] may be an interesting future direction.''

  1. It would be beneficial to provide a rationale for why the Bethe-Salpeter equation is an important future direction.

We thank Referee B for pointing out the insufficient description. Computing dynamical responses of correlated systems has been a challenging issue in condensed matter physics. This requires efficiently solving diagrammatic equations at the two-particle level, including the Bethe-Salpeter equation. In addition, solving the Bethe-Salpeter equation requires matrix multiplications of two-particle objects, which is more challenging than compressing objects. We have revised and extended the second last paragraph in the summary section.

  1. When Figure 9 is mentioned, it would help to provide an explanation for the dip near 1000, which is distinct from all other models presented.

We thank Referee B for the suggestion. The dip is due to a sign change in $G(\tau)$, i.e., the cancellation of contributions from positive and negative coefficients. We have added an explanation where Fig. 9 is mentioned (Appendix A).

``A small dip around $\tau=1000$ originates from a sign change in $G^\mathrm{F}(\tau)$.''

---

## Round 2 · List of Changes

The modifications are highlighted in red in the PDF copy of [the revised manuscript](https://www.dropbox.com/scl/fi/k53smtgds0hwdd8w381vp/qtt_scaling_matsubara_revised.pdf?rlkey=exs1zqoet1aogwgy0rpqot5pb&dl=0).

* Added an explanation on the validity of the models in the abstraction.
* Emphasized the main finding, i.e., the compressibility of imaginary-time propagator in the abstract.
* Cited extensive QTT-based work outside condensed matter physics Refs. 20--23 in the introduction.
* Cited an applied mathematics paper [31] to stimulate future studies.
* Cited Ref. 31 on structure optimization of tree tensor networks.
* Improve the visibility of the colormaps in Figs. 5, 6, 7. 8.
* Made minor revisions throughout the manuscript to improve the readability.

---

## Editorial Decision

published